# Test of the Rehabilitation Goal Screening (ReGoS) Tool to Support Decision Making and Goal Setting in Physical and Rehabilitation Medicine Practice

**DOI:** 10.3390/ijerph192315562

**Published:** 2022-11-23

**Authors:** Christoph Gutenbrunner, Christoph Korallus, Christoph Egen, Joerg Schiller, Christian Sturm, Lidia Teixido, Isabelle Eckhardt, Andrea Boekel

**Affiliations:** Department of Rehabilitation Medicine, Hanover Medicial School, University Hospital in Hanover, 30625 Hannover, Germany

**Keywords:** goal setting, rehabilitation, shared decision-making, ICF, screening

## Abstract

Background: It has already been shown that it is feasible to use International Classification of Functioning, Disability and Health (ICF) Sets as self-assessment instruments. We used this idea to design an ICF-based screening tool to assess patients of a broadly based rehabilitation department. It was developed for the purpose of having a screening tool before taking the anamnesis, as well as for rehabilitation planning and follow-up. Methods and Materials: The Rehabilitation Goal Screening (ReGoS) instrument is a self-report questionnaire which was developed based on the most relevant domains from the ICF Core Sets for chronic pain and rehabilitation. The ICF categories were translated into plain language and 0–10 Likert scales were used. A retrospective analysis of routine clinical data using the ReGoS tool, Work Ability Index (WAI) and Hospital Anxiety and Depression Scale (HADS) in paper- or tablet-based form was performed. Results: The average age of the N = 1.008 respondents was 53.9 years (SD = 16.2). Of the respondents, 66% (*n* = 665) were female. At the time of the survey, 48.3% (*n* = 487) of the patients were employed. ReGoS results demonstrated that the highest restrictions on a scale from 0 to 10 were found in the areas of energy and drive (M = 5.79, SD = 2.575) and activities of daily living (M = 5.54, SD = 2.778). More than a third of the respondents rated their work ability as critical. Conclusion: The use of the ReGoS instrument as an ICF-based screening tool based on a self-report questionnaire provides relevant information for clinical diagnosis, participative goal setting and a detailed functional capacity profile.

## 1. Introduction

Goal setting is an important factor for successful rehabilitation [1]. It can reduce patients’ unrealistic expectations and increase motivation. To some extent, it also reduces the disappointment of patients, health professionals and medical doctors. Last but not least, it may help to avoid an unlimited series of rehabilitation treatments that may occur if the goal has been unrealistic and will never be reached.

One central precondition for goal setting in rehabilitation is that the patient and all team members agree on the goal and define a parameter that can represent successful goal attainment. To achieve this, the patient’s expectations must be integrated with the evidence and/or knowledge-based estimation of the doctor and/or health professional with regard to realistic goals. These two aspects might be contradictory. To solve this problem, a “negotiation” is needed, which can be achieved by using a shared decision-making approach [2]. Factors to be integrated are the results of assessments and functional testing, knowledge about disease mechanisms and individual potential for improvement, as well as information on the effectiveness of interventions.

In outpatient settings, appropriate goal setting is often an extreme challenge because of a number of factors:Lack of time to talk about the details of the patient’s life situation, personal expectations and priorities as well as contextual factors;Lack of unbiased information from the patient (who will be easily influenced by the doctor’s convictions, at least in the first few consultations and before confidence can grow);The fact that sum-scores as outcomes of assessment tools are not useful for goal setting;The fact that besides the patient’s health condition, function-related goals must be taken into account, especially with regard to activities, participation and the modification of contextual factors;The difficulty in reducing multiple expectations to a “leading” goal that can be used for follow-up goal attainment.

In an inpatient setting, a team-integrated approach from the beginning and the possibility to define the goals after an intensive and multidimensional assessment are helpful for defining realistic and agreed-upon goals. In outpatient settings, however, it is often necessary to define the goal on the first day of the rehabilitation process during patient–doctor dialogue. To support this challenging task, we sought to develop a tool called “Rehabilitation Goal Screening” (ReGoS) that can be used (or filled-in) by the patient him- or herself without much support and ideally within a few minutes while waiting for the doctor’s appointment. This tool should be a comprehensive assessment that claims to provide an overview of physical and mental functions, activities of daily living and participation from the patient’s perspective and should be used to facilitate the goal-setting process [3].

Such a tool is supposed to help:Obtain the patients’ own and unbiased perspective before discussion with the MD;Support the negotiation process between patient and MD with regard to goal setting;Support a patient-centered interview and facilitate shared decision making;Support efficient time management in outpatient rehabilitation practices.

The aim of this article is (i) to present the development of the tool, (ii) to show its digitized clinical application in a clinical outpatient setting and its results, and (iii) show the internal consistency of the ReGoS screening instrument (Appendix A).

## 2. Materials and Methods

In this retrospective cohort study using routine clinical data, we show the process of screening tool development, its pathway for use in clinical practice and the application of the tool. We also show descriptive data from the cohort of the Department of Rehabilitation Medicine, where the tool was implemented.

In order to develop any screening tool, it should incorporate the following predefined attributes:The tool should provide information on the most relevant aspects of functioning.The information gathered should reflect the patient’s individual rehabilitation goals.The tool should use plain language so that the patient is able to fill in the questionnaire without any external support.The patient should be given the opportunity to think about and formulate his or her individual goals.The result of the questionnaire should be transferable to a profile that can be interpreted by the doctor at a glance.The results of the screening must be available immediately as a basis for goal setting negotiation.

The idea was transferred into following flowchart (Figure 1).

Additionally, it was considered important to include screening for work ability and mental problems. For the purpose defined above, it is most relevant to produce a functioning profile and to obtain some free-text information about the main problem and the individual rehabilitation goals formulated by the patient. A seemingly feasible way to simplify this for clinical routine is the production of an assessment linked to an 11-step numerical rating scale [4]. Other projects have already shown that it is possible to use the ICF sets for self-assessment [5,6].

The following steps were taken in the tool development process:Use of items from the ICF Core set for rehabilitation [7,8];Selection of not more than 20 items by two experienced PRM physicians according to the criteria, using only one item that can represent a group of categories (e.g., out of washing, dressing and taking care of body parts);Translate the selected categories into questions written in plain language;Use a Likert scale to enable a differentiated graphical profile;Include a question on the patient’s own top-priority rehabilitation goals and ask them to rate these goals on another Likert scale.

In the last two questions of the ReGoS tool, respondents have the opportunity to describe and weigh their main problems and main treatment goals by writing a free-text answer. In addition to the selected ICF categories, two standardized screening tools were added: the Work-Ability Index (WAI), to look at the patient’s potential to stay at or return to work, and the Hospital Anxiety and Depression Scale (HADS), to obtain an estimate for mental problems [9,10,11]. After developing the final version, the questionnaire was prepared to be filled-in on a tablet computer and a profile was generated for the doctor’s desktop computer.

### 2.1. Case Study as Practical Example

In a recent case study, we dealt with a 70-year-old female patient with fibromyalgia syndrome. We saw strong limitations in different items of the questionnaire (Figure 2). Through the use of our screening tool, it became clear how diverse the restrictions were. It was typical to observe the incomprehensibly high pain intensity of 10/10 on the numeric rating scale (NRS), which shows a disturbed perception of pain. In the personal anamnesis interview, it could be inferred that the restriction of sleep quality was especially stressful. Therefore, this was defined as the first rehabilitation goal. Appropriate therapies were discussed and prescribed. A further assessment after completion of the first therapy series can show the effectiveness of the treatment in relation to the therapy goal, and new goals, preferably on the level of activities and participation, can be defined.

### 2.2. Participants and Setting

The sample included outpatients of the Department of Rehabilitation Medicine who had an initial or re-presentation appointment during the period from 4 January 2017 to 29 November 2019.

At presentation during patient registration, patients were provided with a paper questionnaire or a tablet PC with the questionnaire for immediate completion, depending on their preference. In order to avoid delays, all patients were called in 15 min earlier. If necessary, administrative staff members supported the patients as they completed the questionnaire. The questionnaire was completed prior to the doctor’s appointment so that the results could be used in planning rehabilitation goals and strategies.

#### 2.2.1. Subjective Work Ability

The participants’ subjective work ability was assessed using the German version of the Work Ability Index (WAI) on an eleven-point scale (11–12). Via scores ranging from 7 to 49, the WAI mapped whether the participant’s current work ability should be classified as critical (7–27 points), moderate (28–36 points), good (37–43 points) or very good (44–49 points). According to the score, the participant’s personal goal can be characterized as an intention to restore, improve, support or maintain their ability to work [12].

#### 2.2.2. Mental Condition

The German version of the Hospital Anxiety and Depression Scale (HADS) self-assessment procedure consists of seven anxiety and seven depression questions, and can be used as a screening instrument and also to assess the patient’s course [10,13]. The Cronbach’s alpha reliabilities and split-half reliabilities for the two subscales were both 0.80. The retest reliability was 0.80 after two weeks and 0.70 after six months. Sensitivity and specificity for case detection were approximately 0.80 [10]. To form the raw scores of the subscales, the scores obtained per item were added scale by scale. The score values were divided as follows:-<7 = unremarkable;-8–10 = borderline;-11–14 = severe symptomatology;-15–21 = very severe symptomatology.

### 2.3. Statistical Analysis

Descriptive analysis was performed using means, standard deviations (SD), percentages and frequencies calculated from sociodemographic data and from the results of the HADS, WAI and ReGoS instruments. Cronbach’s alpha was calculated separately to these subscales to examine the internal consistency of the ReGoS instrument.

## 3. Results

### 3.1. Sociodemographic Data

The data retrieved from the ReGoS screening contained 1008 records after the first set of interviews. The mean age of the respondents was 53.9 years (SD = 16.2). Respondents ranged in age from 13 to 92 years. Respondents were 66% (*n* = 665) female. The percentage of respondents employed at the time of the screening was 48.3% (*n* = 487) (Table 1).

### 3.2. Anxiety and Depressiveness (HADS)

HADS scores were available for *n* = 680 respondents. On average, respondents scored 1.81 for anxiety. More than half had unremarkable scores, while 17.8% had severe and 7.9% very severe symptoms (Figure 2).

In the area of depressiveness, the mean score was 1.65, with scores averaging in the “unremarkable to borderline” range. Nearly two-thirds showed unremarkable values, with 14.4% showing severe symptoms and 6.9% very severe symptoms (Figure 3).

### 3.3. Work Ability (WAI)

Data from *n* = 285 (28.3% of 1008) people were available for the WAI. More than one-third of the respondents rated their ability to work as critical (36.8%). Another third was classified as moderate by the WAI (32.6%). A good or very good work ability index was achieved by 24.6% or 6%, respectively.

### 3.4. Main Diagnoses (ICD)

Of the main diagnoses in the group of patients, 62.1% were related to musculoskeletal disorders, followed by mental and behavioral disorders at 14.3%. Patients with diseases of the ear (6.2%) and the nervous system (3.9%) were also present.

### 3.5. Detailed Functioning Profile and Free Texts of the Study Cohort

The areas of life energy and drive (M = 5.8, SD = 2.6) and carrying out daily routine (M = 5.5, SD = 2.8) showed the highest restrictions on average. This was followed by disease-related restrictions that limit the ability to recover or pursue a leisure activity (M = 4.9, SD = 2.9), impairment of sleep functions (M = 4.8, SD = 3.1), emotional functions (M = 4.9, SD = 2.9) and impairments in dealing with stress and mental demands (M = 4.4, SD = 3.1). Restrictions when changing basic body positions (M = 3.9, SD = 3.2), walking (M = 4.0, SD = 3.4) and performing remunerative or voluntary activities (M = 4.2, SD = 3.6) were, on average, rated at 4 out of 10 points. The average rating for restrictions on doing housework (M = 3.4, SD = 2.9) was 3 out of 10 points. Restrictions in using transportation were rated rather lower (M = 1.9, SD = 2.7). Activities such as dressing (M = 1.1, SD = 1.9), washing (M = 0.9, SD = 1.9), using the toilet (M = 0.7, SD = 1.6) or eating (M = 0.6, SD = 1.6) were relatively less restricted (Figure 4).

Using the free-text options, patients described their most limited function or activity and also indicated which function or activity was personally their most important treatment goal. They also indicated how severe this limitation felt on a scale from 0 to 10. Table 2 shows some examples of this.

### 3.6. Rehabilitation Goals

With more than 10 percent each, “Carrying out daily routine”, “Sensation of pain”, “Walking” and “Energy and drive functions” were used particularly frequently as rehabilitation goals, while “Eating”, “Toileting” and “Washing oneself”, for example, were defined as goals in less than one percent of the cases (Figure 5). In addition to these ICF-based goals, the patients were able to add free-text mentions of their most severe limitation and of the limitations to their most important goal. The patient’s most severe limitation was additionally defined as a therapy goal 713 times and limitations to the most important goal were reported 646 times.

### 3.7. Internal Consistency of ReGoS

A Cronbach’s alpha of 0.91 shows that the instrument has excellent internal consistency.

## 4. Discussion

Soon after the publication of the International Classification of Functioning, Disability and Health (ICF), the issue of the feasibility of the application was addressed [14,15]. A milestone was the development of ICF core sets for specific health conditions and rehabilitation settings [16]. Later on, core sets (or minimum reporting sets) for reporting on a system and services level as well as for rehabilitation quality management were developed [15]. For practical use in a number of languages, so-called “simple intuitive descriptions” for the generic and rehabilitation sets were developed [17,18], facilitating the use of these sets in a broad range of settings. However, the use of ICF core sets in routine clinical practice has remained limited and has been mainly restricted to institutions with multi-professional teams and to inpatient settings. Arguments for not using the ICF in daily clinical practice include the fact that health professionals have a limited ability to evaluate the patient and that the number of items, even in the core sets, is still too high.

For that reason, a pragmatic clinical approach has been taken in the outpatient section of the Department of Rehabilitation Medicine of Hannover Medical School.

Thus, the developed tool is not an assessment instrument for scoring. It was designed as a screening questionnaire to generate an individual patient functioning profile to be used as a facilitator in the doctor’s clinical evaluation. It is used to screen the patient’s main problems and provides useful information to give the doctor a comprehensive insight into the patient. The instrument is not intended to replace well-established assessments, but to provide a clear representation of the patient’s subjective problems and to support rehabilitation goal setting in a patient-centered shared decision-making process.

The results of this study showed that this is a feasible approach, and the questionnaire could be filled in by the patients without relevant problems. The quality of the screening tool is corroborrated when comparing the results with conventional established questionnaires.

Our item on occupational limitations correlates strongly with the WAI score (r = 0.709, *p* < 0.0001, *n* = 284). Our item on emotional functioning also correlates strongly with both HADS scores (HADS Anxiety, r = 0.594, *p* < 0.0001; HADS Depression, r = 0.581, *p* > 0.0001; *n* = 680).

However, test-retest reliability, as well as validity and sensitivity, have not been tested yet. For the purpose of pre-screening in advance of routine diagnostic procedures, this seems to be acceptable.

The profile of the evaluated patients showed that the work ability of one-third of the studied group was critical and more than 60% of the diagnoses were related to diseases of the musculoskeletal system and connective tissue. Interestingly, this proportion serves as a reminder of the global burden of disability and rehabilitation needs worldwide, as recently published by Cieza et al. [19].

After the extensive test phase of the screening instrument, we can, on one hand, make statements about the clientele of an outpatient department of rehabilitation medicine, but on the other hand, we can also draw conclusions about the requirements for the employees. We see a wide spectrum of diseases for which the field of physical and rehabilitative medicine is already known [20]. A total of 76.4% of our main diagnoses were related to musculoskeletal and psychosomatic diseases. These diseases are of high relevance in Germany and account for approximately 30% of the healthy life-years (DALYs) lost due to limitations [21].

On the “Depressiveness” section of the HADS scale, we measured conspicuous values in 21.3% of the participants, indicating severe to very severe symptomatology. On the “Anxiety” section of the scale, the figure was as high as 25.7%. In the normal German population, 5.2% of men and 8.1% of women have conspicuous values for the anxiety scale of the HADS. For the depression scale, abnormal scores are reported in 9.6% of men and 9.3% of women, respectively, in the normal population [22]. Thus, our sample scored significantly higher than the national average.

Looking at the work ability of our study population, parallels to studies that have investigated work ability in patients with chronic pain were identified [23,24,25]. Even chronic health problems in general affect work ability in ways we have also measured [26].

When looking at the limitations in the ICF categories, among the greatest limitations a significant proportion of dimensions belong to the group of items related to mental health. Classical categories concerning the musculoskeletal system are more in the background than the distribution of diagnoses would suggest. The goals set together with the patient again reflect this well. This implies a strong integration of mental diagnosis and therapy also present in the outpatient rehabilitative setting.

Further considerations about ICF-based goal setting relate to the high proportion of free-text mentions used for goal setting. The question must be raised whether the language of the ICF is also suitable for communication between physician and patient, or whether it is simply easier to translate the most significant concerns stated in advance by the patient into goals, rather than using higher-level goals.

From a clinical perspective, the unbiased patient’s perspective, not influenced by any professional assistance, provided by our compact ICF set is very useful. Compared to conventional assessments, it is time-saving and provides a good overview of many areas of life. The information obtained can be quickly overviewed by the practitioner through the profile created (Figure 2). In addition, we can confirm the suitability of our ICF items for goal setting, but we need to discuss whether a multilevel, rehabilitation phase-specific adaptation of the goal setting tool may be necessary.

So far, we have been able to evaluate a feasible and useful ICF-based screening tool for use in the context of outpatient rehabilitation. This has been shown in a diverse and, above all, large group of patients. Based on the subjective amount of feedback from the patients, we can say that it is easy to be filled in. The scientific quality criteria achieved are also extremely positive. A Cronbach’s alpha of 0.91 indicates a reliable scale, so we would view the ReGoS tool as reliable [27].

One limitation of our study is the lack of information on the patients’ socioeconomic context, which we did not include in the ReGoS instrument, so we cannot provide absolute details of the sample at this time. In the future, we would also like to present the progress of the assessment over further measurement periods and would like to see the practicality of using it with inpatients. Another limitation of our work is that HADS and WAI were added to the assessment at a later stage, so we could not demonstrate the complete data here.

## 5. Conclusions

A self-reporting ICF-based tool for screening and goal setting is easy to implement, easy to use, time-efficient even with large patient volumes and allows pragmatic use of the ICF in everyday clinical practice, with favorable statistical characteristics. This concept can be adapted for other patient groups.

## Figures and Tables

**Figure 1 ijerph-19-15562-f001:**
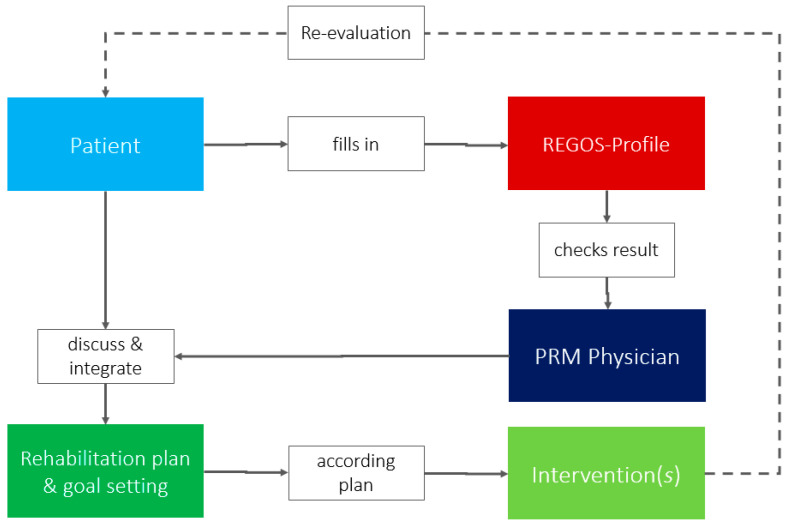
Pathway of REGOS tool use in clinical practice.

**Figure 2 ijerph-19-15562-f002:**
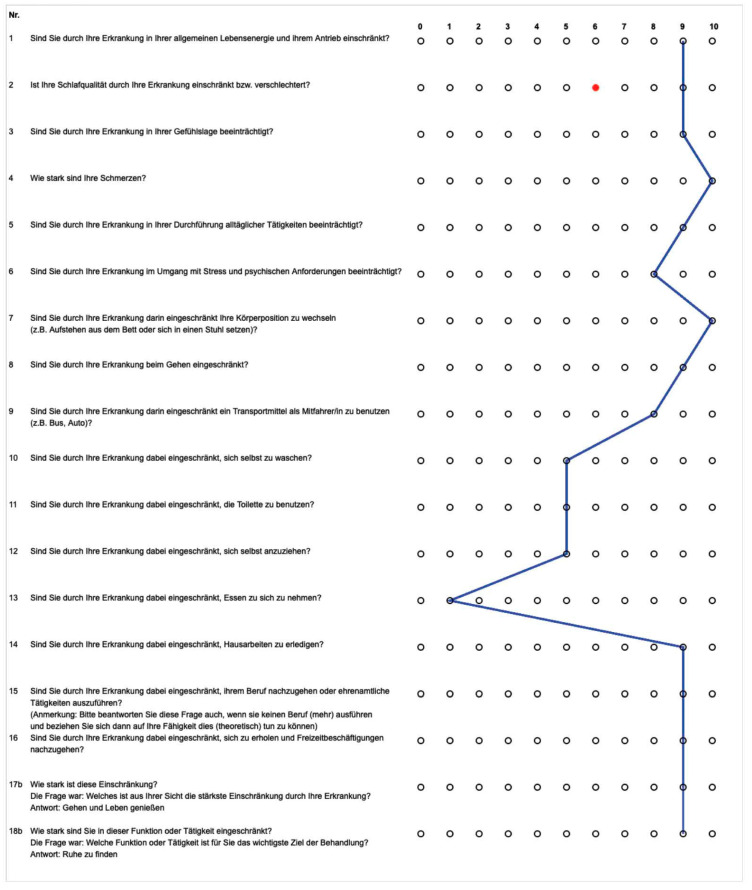
Case study of a patient showing the profile of limitations (**blue line**) and rehabilitation goal (**red dot**).

**Figure 3 ijerph-19-15562-f003:**
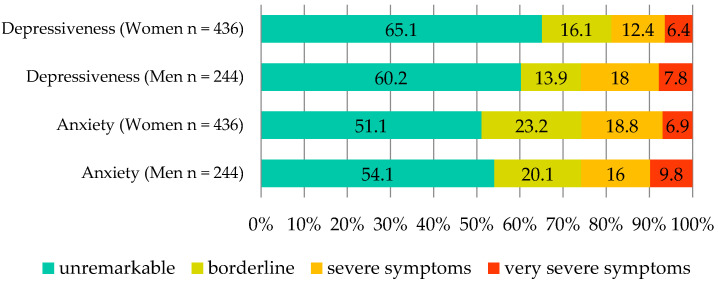
Frequency of the Hospital Anxiety and Depression scale (HADS) depression and anxiety scores by gender in %.

**Figure 4 ijerph-19-15562-f004:**
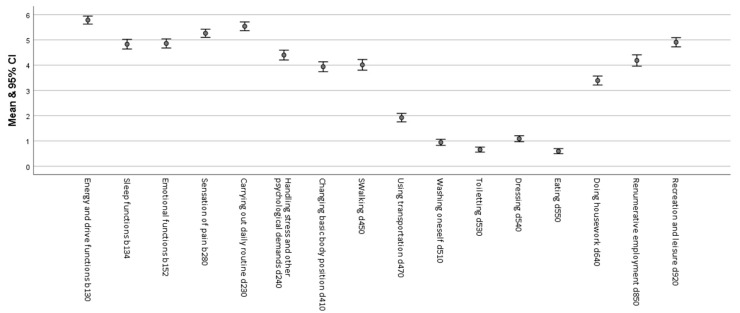
Detailed functioning profile obtained from the ReGoS tool.

**Figure 5 ijerph-19-15562-f005:**
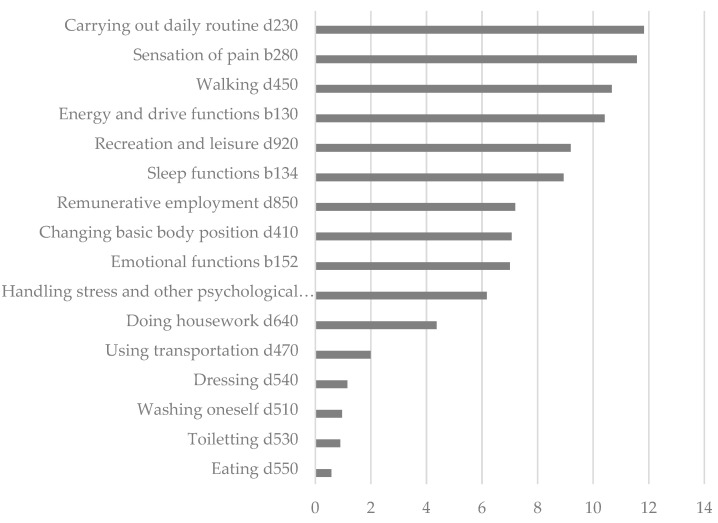
Frequencies of the selected ICF-based rehabilitation goals (%) (*n* = 1556).

**Table 1 ijerph-19-15562-t001:** Sociodemographic sample characteristics. Abbreviations: M, mean; SD, standard deviation.

Sociodemographic Sample Characteristics	N = 1.008
Age in years, M (SD)	53.9 (16.2)
Gender, *n* (%)	
Female	665 (66.0)
Male	343 (34.0)
Employed, *n* (%)	
Yes	487 (48.3)
No	521 (51.7)

**Table 2 ijerph-19-15562-t002:** Patients’ free-text answers about their limitations of function and activity as well as their most important treatment goal, including a rating of their severity on a scale from 0 = no problem to 10 = maximal problem.

Patient’s ID	Function or Activity with the Greatest Limitation	Severity	Most Important Treatment Goal (Function or Activity)	Severity
ID xxxxxx8310	“walking and standing”	8	“To be able to walk properly again. I have the feeling that I am not moving forward properly.”	8
ID xxxxxx0420	“Lack of strength in the hands. Fine motor skills severely limited.”	5	“Avoid further ulnar deviation”	5
ID xxxxxx5220	“Many trips to the toilet during activities and mental problems. Infection of the urinary tract.”	3	“Control of bladder function during urges.”	3

## Data Availability

Not applicable.

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
