# Peer review of "Test of the Rehabilitation Goal Screening (ReGoS) Tool to Support Decision Making and Goal Setting in Physical and Rehabilitation Medicine Practice"

_ijerph, 2022, doi:10.3390/ijerph192315562_

Round 1

Reviewer 1 Report

Very Respected Authors,

After carefully reading your manuscript I have few suggestions. IThe Abstract is well structurated but I suggest you to avoid the abbrevations or if they are  neccesarry you have to write the full name of a term and in a brackets abbrevation. Ethical approvement is usually stands at the end of the section Material and Methods.  I suggest you to write the Type of a Study . You wrote that you retospectively analyzed routine clinical data. My question is  is that was retrospective cohort study or desriptive, or case-control study

Author Response

Dear Reviewer,

We would like to thank you for the helpful comments to improve our manuscript. Please find a point-to-point reply of your comments below.

  1. The Abstract is well structurated but I suggest you to avoid the abbrevations or if they are neccesarry you have to write the full name of a term and in a brackets abbrevation.
  • We wrote the full name and added the abbreviation in brackets.

  1. Ethical approvement is usually stands at the end of the section Material and Methods.
  • Due to the journals instructions for authors the Institutional Review Board Statement and approval number for studies should be written in the section Institutional Review Board Statement. Please see line 446.

  1. I suggest you to write the Type of a Study . You wrote that you retospectively analyzed routine clinical data. My question is  is that was retrospective cohort study or desriptive, or case-control study
  • We added the type of study (line 85ff).

Reviewer 2 Report

The topic raised is very interesting, as assessing goals in rehabilitation is difficult because patients have comorbidities in addition to musculoskeletal disorders.

The authors should shorten the content of the manuscript and, present what is most important in a goals evaluation.

Author Response

Dear Reviewer,

We would like to thank you for the helpful comments to improve our manuscript. Please find a point-to-point reply of your comments below.

  1. All abbreviations used in the manuscript should have their full name, given the first time.
  • We wrote the full name and added the abbreviation in brackets.

  1. Verse 30 - as well as in the all maunskrypt as a whole - which therapists do the authors write about?
  • We replaced the term „therapist“ with the term „health professional“ which includes occupational therapists, speech and language therapists, physiotherapists etc.

  1. The authors state the purpose of the work followed by a case study – why? What value does this have for the manuscript?
  • In order to give an example of clinical application of the ReGoS, we added the case study. To shorten the manuscript, we now deleted figure 7 as another example of an individual profile.

  1. The material and methods presented depart from the aim of the work - the authors should sort it out.
  • In order to give the readers and maybe future users of the screening tool an idea of the application, we also need to show the demographic data and results of the tool, so they can evaluate, if their cohort in their department or clinic could also be screened with this tool. Due to this, we added the presentation of the results in the aims (line 80).

  1. The material and methods – „The patient should be able to fill in the questionnaire without any external support. The patient should be given the opportunity to think about and formulate his or her individual goals.”, and how does this relate to patients who cannot identify, describe, write, complete a questionnaire? The authors should state to which patients this is addressed
  • In developing the screening tool, attributes were defined that it should fulfill, including that patients should be able to complete the questionnaire as independently as possible. The intention behind this was to use simple language that could be understood by medical laypersons. We added this aspect in the text (line 93)

  1. The authors should refrain from including all figures e.g. figure 4 and 5.
  • We have removed the figures.

  1. The authors should shorten the content of the manuscript and, present what is most important in a goals evaluation
  • We shortened the content of the manuscript. Please see the changes in sections material and methods and results the manuscript. We would also add the ReGoS-Instrument not in the text but as a download.

Round 2

Reviewer 2 Report

The authors have revised the manuscript.

Author Response

Dear reviewer,

thank you for reviewing our manuscript.

We included line 425 Limitations as a separate item.

Best regards from the author team
